# SOLAR: DEEP STRUCTURED REPRESENTATIONS FOR MODEL-BASED REINFORCEMENT LEARNING

## ABSTRACT

Model-based reinforcement learning (RL) methods can be broadly categorized as global model methods, which depend on learning models that provide sensible predictions in a wide range of states, or local model methods, which iteratively refit simple models that are used for policy improvement. While predicting future states that will result from the current actions is difficult, local model methods only attempt to understand system dynamics in the neighborhood of the current policy, making it possible to produce local improvements without ever learning to predict accurately far into the future. The main idea in this paper is that we can learn representations that make it easy to retrospectively infer simple dynamics given the data from the current policy, thus enabling local models to be used for policy learning in complex systems. We evaluate our approach against other model-based and model-free RL methods on a suite of robotics tasks, including a manipulation task on a real Sawyer robotic arm directly from camera images.

## 1 INTRODUCTION

Model-based reinforcement learning (RL) methods use learned models in a variety of ways, such as planning (Levine & Abbeel, 2014; Deisenroth et al., 2014) and generating synthetic experience (Sutton, 1990). We can categorize model-based algorithms as either global model methods, where models are used for planning and trained to give accurate predictions for a wide range of states, or local model methods, where simple models provide gradient directions that are used for policy improvement. On simple, low-dimensional tasks, model-based approaches have demonstrated remarkable data efficiency, learning policies for systems like cart-pole swing-up with under 30 seconds of experience (Deisenroth et al., 2014; Moldovan et al., 2015). However, for more complex systems, one of the main difficulties in applying model-based methods is model bias: local models will often underfit complex systems, but may still be preferred over global models which tend to overfit in the low-data regime and may be difficult to incorporate into control methods (Deisenroth et al., 2014).

Most global model methods use the model to make forward predictions and then backpropagate through those predictions. However, this places a heavy burden on the dynamics model, and forward prediction often suffers from significant drift over longer trajectories. In contrast, local models are typically only used to provide gradient directions for local policy improvement (Levine & Abbeel, 2014), and thus a common choice for local model methods is to use linear models, which can themselves be interpreted as gradients. As illustrated in Figure 1, in our work, we present a method that automatically encourages learning representations where linear models better fit the data. From this, we devise an efficient local model method based on the linear-quadratic regulator (LQR) (Camacho & Bordons, 1997; Todorov & Li, 2005; Levine & Abbeel, 2014) that utilizes linear models for gradient directions for policy improvement. Our motivation is similar to that of Watter et al. (2015); Finn et al. (2016); however, as discussed in section 5, our representation learning method specifically allows us to construct a local model method that performs inference in the latent space in order to improve the policy, rather than focusing on forward prediction and planning.

Our main contribution is a representation learning and model-based RL procedure, which we term stochastic optimal control with latent representations (SOLAR), which jointly optimizes a latent representation and model such that inference produces local linear models that provide good gradient directions for policy improvement. We demonstrate empirically in section 6 that SOLAR is able to learn policies directly from raw, high-dimensional observations in several robotic environments

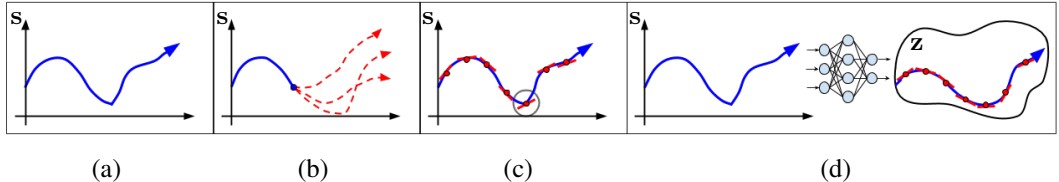

Figure 1: (a) A pictoral depiction of a trajectory for a one-dimensional system. (b) Global models may be used for prediction or planning forward through time, as depicted in red, but this can suffer from trajectory drift for complex systems. (c) Local linear models are fit to trajectories and do not suffer from drift, but may fit the system poorly for complicated interactions such as contacts, as illustrated by the poor model fit circled in gray. (d) Our method finds an embedding of observed trajectories into a latent space where local linear models produce a better fit.

including a simulated nonholonomic car, a simulated two degree-of-freedom (DoF) arm, and a real 7-DoF Sawyer arm, all of which are learned directly from image pixels. We compare to existing state-of-the-art RL methods and show that SOLAR, while significantly more data efficient than model-free methods, exhibits superior performance compared to other model-based methods.

## 2 PRELIMINARIES

We first formalize our problem setting as a Markov decision process (MDP) $M = (\mathcal{S}, \mathcal{A}, p, C, \rho, T)$, where the state space $\mathcal{S}$, action space $\mathcal{A}$, and horizon $T$ are known, but the dynamics function $p(\mathbf{s}_{t+1}|\mathbf{s}_t, \mathbf{a}_t)$, cost function $C(\mathbf{s}_t, \mathbf{a}_t)$, and initial state distribution $\rho(\mathbf{s}_0)$ are unknown. The goal of reinforcement learning is to optimize a policy $\pi(\mathbf{a}_t|\mathbf{s}_t)$ to minimize the expected sum of costs $\eta[\pi] = \mathbb{E}_{\pi,p,\rho}\left[\sum_{t=0}^{T} C(\mathbf{s}_t, \mathbf{a}_t)\right]$ under the distribution induced by the initial state distribution, dynamics function, and policy. Model-based methods decompose this problem into policy and model optimization subproblems, and we discuss each subproblem as it relates to our approach.

### 2.1 MODEL-BASED POLICY SEARCH

Policy search methods directly optimize parameterized policies with respect to $\eta(\theta) \triangleq \eta[\pi_\theta]$ where the parameters $\theta$ may be, for example, weights in a neural network or matrices for a linear policy. Model-based policy search methods typically build models $\left(\hat{\rho}, \hat{p}, \hat{C}\right)$ of the unknown quantities and compute the gradient of $\hat{\eta}(\theta) \triangleq \mathbb{E}_{\pi_\theta, \hat{p}, \hat{\rho}}\left[\sum_{t=0}^{T} \hat{C}(\mathbf{s}_t, \mathbf{a}_t)\right]$ with this model. One particularly tractable model is the linear-quadratic system (LQS), which models the initial state distribution as Gaussian, the dynamics as time-varying linear-Gaussian (TVLG), and the cost as quadratic, i.e.,

$$\hat{p}(\mathbf{s}_{t+1}|\mathbf{s}_t, \mathbf{a}_t) = \mathcal{N}\left(\mathbf{s}_{t+1} \,\middle|\, \mathbf{F}_t \begin{bmatrix} \mathbf{s}_t \\ \mathbf{a}_t \end{bmatrix}, \Sigma_t\right), \quad \hat{C}(\mathbf{s}_t, \mathbf{a}_t) = \frac{1}{2} \begin{bmatrix} \mathbf{s}_t \\ \mathbf{a}_t \end{bmatrix}^\top \mathbf{C} \begin{bmatrix} \mathbf{s}_t \\ \mathbf{a}_t \end{bmatrix} + \mathbf{c}^\top \begin{bmatrix} \mathbf{s}_t \\ \mathbf{a}_t \end{bmatrix}. \quad (1)$$

Any deterministic policy operating in an environment with smooth dynamics can be locally modeled with a time-varying LQS (Boyd & Vandenberghe, 2004), while low-entropy stochastic policies are modeled approximately. This makes the time-varying LQS a reasonable local model for many dynamical systems. Furthermore, the optimal policy at any time step given the model is a linear function of the state and the optimal maximum-entropy policy is linear-Gaussian (Tassa et al., 2012; Levine & Koltun, 2013). As shown in Jacobson & Mayne (1970); Todorov & Li (2005), these optimal policies can be computed in closed form using dynamic programming by computing the first and second derivatives of the Q (cost-to-go) and value functions:

$$Q_{\tilde{\mathbf{s}},t} = \mathbf{c}_{\tilde{\mathbf{s}},t} + \mathbf{F}_{\tilde{\mathbf{s}},t}^\top V_{\mathbf{s},t+1}, \qquad Q_{\tilde{\mathbf{s}}\tilde{\mathbf{s}},t} = \mathbf{C}_{\tilde{\mathbf{s}}\tilde{\mathbf{s}},t} + \mathbf{F}_{\tilde{\mathbf{s}}\tilde{\mathbf{s}},t}^\top V_{\mathbf{s}\mathbf{s},t+1}\mathbf{F}_{\tilde{\mathbf{s}}\tilde{\mathbf{s}},t},$$

$$V_{\mathbf{s},t} = Q_{\mathbf{s},t} - Q_{\mathbf{s}\mathbf{a},t}Q_{\mathbf{a}\mathbf{a},t}^{-1}Q_{\mathbf{a},t}, \quad V_{\mathbf{s}\mathbf{s},t} = Q_{\mathbf{s}\mathbf{s},t} - Q_{\mathbf{s}\mathbf{a},t}Q_{\mathbf{a}\mathbf{a},t}^{-1}Q_{\mathbf{a}\mathbf{s},t}.$$

Here, similar to Tassa et al. (2012), we use subscripts to denote derivatives, and we use $\tilde{\mathbf{s}}$ to abbreviate $\begin{bmatrix} \mathbf{s} \\ \mathbf{a} \end{bmatrix}$. Once these values are computed, the optimal maximum-entropy policy is TVLG, i.e.,

$$\pi_\theta(\mathbf{a}_t|\mathbf{s}_t) = \mathcal{N}\left(\mathbf{K}_t\mathbf{s}_t + \mathbf{k}_t, \mathbf{S}_t\right), \text{ where } \mathbf{K}_t = -Q_{\mathbf{a}\mathbf{a},t}^{-1}Q_{\mathbf{a}\mathbf{s},t}, \mathbf{k}_t = -Q_{\mathbf{a}\mathbf{a},t}^{-1}Q_{\mathbf{a},t}, \mathbf{S}_t = -Q_{\mathbf{a}\mathbf{a},t}^{-1}.$$

We refer the reader to Appendix A and Levine & Abbeel (2014) for further details. Prior work assumes access to a compact, low-dimensional state representation (Deisenroth et al., 2014; Levine & Abbeel, 2014; Nagabandi et al., 2018), and as we show in section 6, this precludes these local model methods from operating on complex observations such as images. In subsection 2.2 and section 3, we describe a probabilistic latent variable model and variational inference procedure that, conditioned on a full trajectory of observations, produces local models that can be used for policy improvement, enabling us to utilize this local model method in image-based domains.

## 2.2 LEARNING LATENT DYNAMICS MODELS

The local model-based method described above requires us to learn both a quadratic cost function as well as a linear dynamical system (LDS). We utilize the Bayesian LDS model, which is given by

$$\mu_{\hat{\rho}}, \Sigma_{\hat{\rho}} \sim \mathcal{NIW}(\Psi, \nu, \mu_0, \kappa), \quad \mathbf{F}_t, \Sigma_t \sim \mathcal{MNIW}(\Psi, \nu, \boldsymbol{M}_0, \boldsymbol{V}) \text{ for } t \in [0, \ldots, T-1],$$

$$\mathbf{s}_0 \mid \mu_{\hat{\rho}}, \Sigma_{\hat{\rho}} \sim \mathcal{N}(\mu_{\hat{\rho}}, \Sigma_{\hat{\rho}}), \qquad \mathbf{s}_{t+1} \mid \mathbf{s}_t, \mathbf{a}_t \sim \mathcal{N}\left(\mathbf{F}_t \begin{bmatrix} \mathbf{s}_t \\ \mathbf{a}_t \end{bmatrix}, \Sigma_t\right) \text{ for } t \in [0, \ldots, T-1],$$

Where $\mathcal{NIW}$ is the normal-inverse-Wishart distribution and $\mathcal{MNIW}$ is the matrix normal-inverse-Wishart (MNIW) distribution. This probabilistic graphical model (PGM) allows for tractable approximate inference, i.e., Bayesian linear regression, and also captures uncertainty in the form of a posterior distribution over the initial state and dynamics. However, for dynamical systems with complex non-linear dynamics, this model still suffers from significant bias.

Even when the system is poorly modeled by an LDS in the state space, we might be able to find a latent embedding and model the system as approximately linear in that latent space, which may allow us to find a better-performing policy that operates in the learned latent space. This shifts our problem setting to that of a partially observed MDP, as we do not observe the latent state. In particular, our modeling assumption is that we receive an observation as generated from an underlying unobserved state, and as discussed in section 3, we address this by training a recognition model to infer the latent state. In our experiments in section 6, we provide several observations to our recognition model in order to infer information that cannot be observed from a single observation, such as velocity. We can jointly train an embedding and model using the SVAE framework (Johnson et al., 2016), which allows us to combine arbitrary embedding functions, such as neural networks, with PGMs. The model we build off of is a version of the LDS SVAE presented in Johnson et al. (2016) and is given by

$$\mu_{\hat{\rho}}, \Sigma_{\hat{\rho}} \sim \mathcal{NIW}(\Psi, \nu, \mu_0, \kappa), \quad \mathbf{F}_t, \Sigma_t \sim \mathcal{MNIW}(\Psi, \nu, \boldsymbol{M}_0, \boldsymbol{V}) \text{ for } t \in [0, \ldots, T-1], \quad (2)$$

$$\mathbf{z}_0 \mid \mu_{\hat{\rho}}, \Sigma_{\hat{\rho}} \sim \mathcal{N}(\mu_{\hat{\rho}}, \Sigma_{\hat{\rho}}), \qquad \mathbf{z}_{t+1} \mid \mathbf{z}_t, \mathbf{a}_t \sim \mathcal{N}\left(\mathbf{F}_t \begin{bmatrix} \mathbf{z}_t \\ \mathbf{a}_t \end{bmatrix}, \Sigma_t\right) \text{ for } t \in [0, \ldots, T-1], \quad (3)$$

$$\mathbf{s}_t \mid \mathbf{z}_t \sim f_\gamma(\mathbf{z}_t) \text{ for } t \in [0, \ldots, T], \quad (4)$$

Where $f_\gamma(\mathbf{z})$ is an observation model, parameterized by neural network weights $\gamma$, that outputs a distribution over $\mathbf{s}$, e.g., Gaussian or Bernoulli, depending on the nature of the data. This is very similar to the Bayesian LDS, except we are learning the PGM in the latent space.

Though this model does not admit the same efficient approximate inference algorithms when $f_\gamma$ is nonlinear, an efficient variational inference algorithm has previously been derived by Johnson et al. (2016). We describe the relevant aspects of this algorithm in the next section.

## 3 LEARNING AND MODELING THE LATENT SPACE

In this section, we describe how we extend the LDS SVAE for model-based RL, such that we learn an action-conditioned LQS model in the latent space. This then enables a local model method that can leverage the LQS to infer the dynamics of sampled trajectories. In this way, our model-based RL algorithm circumvents the need for forward prediction, in contrast to model-based RL methods that use model-based rollouts or planning (Nagabandi et al., 2018; Deisenroth et al., 2014). In section 4, we describe how these components are combined into our final method, SOLAR.

Our goal with this model is to learn a latent representation of the state and a prior over the dynamics in this latent representation that is suitable for fitting local dynamics models via posterior inference. Specifically, we are interested in the setting where we have access to trajectories of the form

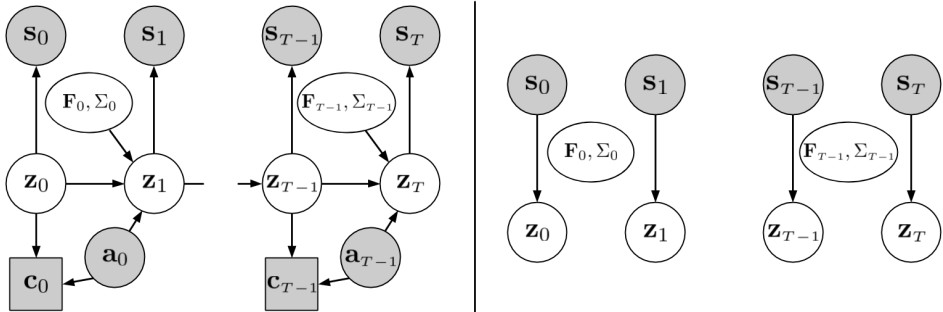

Figure 2: Left: The LQS graphical model. Distributions for each node are as specified in Equation 2-Equation 4, with additional deterministic nodes for observed costs. Right: The variational family we use for our model learning algorithm, with distributions given in Equation 5.

$[\mathbf{s}_0, \mathbf{a}_0, c_0, \ldots, \mathbf{s}_{T-1}, \mathbf{a}_{T-1}, c_{T-1}, \mathbf{s}_T]$, sampled from the system using our current policy and set of previous policies. Our aim is to infer local linear dynamics in the neighborhood of these trajectories, and we learn a model that makes this fitting process more accurate for the observed trajectories, thus enabling our local model method to find good directions for policy improvement.

We build upon the variational inference algorithm presented in Johnson et al. (2016), such that we are maximizing, with respect to both the PGM and neural network parameters, the variational lower bound (ELBO) of our observed data. This algorithm requires variational factors of the form

$$q(\mathbf{z}_t \mid \mathbf{s}_t) = \mathcal{N}\left(e_\phi\left(\mathbf{s}_t\right)\right), \quad q(\mathbf{F}_t, \Sigma_t) = \mathcal{MNIW}(\Psi_t', \nu_t', \boldsymbol{M}_{0t}', \boldsymbol{V}_t') \text{ for } t \in [0, \ldots, T-1]. \quad (5)$$

$e_\phi(\mathbf{s})$ is a recognition model, parameterized by neural network weights $\phi$, that outputs the mean and diagonal covariance of a Gaussian distribution over $\mathbf{z}$. This recognition model is identical to that used in Kingma & Welling (2014); Rezende et al. (2014); Gao et al. (2016), however, as with prior work in the LDS SVAE, we also have variational factors of the form $q(\mathbf{F}_t, \Sigma_t)$, which represent our posterior belief about the system dynamics after observing the collected data. We also model this distribution as MNIW but with updated parameters compared to the prior from Equation 2. Given this, we can formulate the variational lower bound (ELBO) which is given by

$$\begin{aligned}
\mathcal{L} &= \mathbb{E}_q\left[\log \frac{p\left(\{\mathbf{F}, \Sigma\}_{t=0}^{T-1}, \{\mathbf{s}_t\}_{t=0}^{T}, \{\mathbf{a}_t\}_{t=0}^{T-1}, \mathbf{z}_t\}_{t=0}^{T}\right)}{q(\{\mathbf{F}_t, \Sigma_t\}_{t=0}^{T-1}, \{\mathbf{z}_t\}_{t=0}^{T}|\{\mathbf{s}_t\}_{t=0}^{T})}\right] \\
&= \mathbb{E}_q\left[\log \left(\prod_{t=0}^{T} p_\gamma(\mathbf{s}_t|\mathbf{z}_t)\right)\right] \\
&\quad - \sum_{t=0}^{T-1} \mathrm{KL}\left(q(\mathbf{F}_t, \Sigma_t)\|p(\mathbf{F}, \Sigma)\right) - \sum_{t=1}^{T} \mathbb{E}_q\left[\mathrm{KL}\left(q_\phi(\mathbf{z}_t|\mathbf{s}_t)\|p(\mathbf{z}_t|\mathbf{z}_{t-1}, \mathbf{a}_{t-1}, \mathbf{F}_t, \Sigma_t)\right)\right].
\end{aligned}$$

Prior work has shown that, for conjugate exponential models such as the Bayesian LDS, the variational model parameters can be updated using natural gradients, which can be computed in closed form using the variational message passing framework (Winn & Bishop, 2005). Specifically, letting $\lambda$ denote the MNIW parameters of the variational factors on $\{\mathbf{F}_t, \Sigma_t\}_t$, the natural gradient update is

$$\tilde{\nabla}_\lambda \mathcal{L} = \lambda^0 + B\mathbb{E}_q\left[t_{\mathbf{F},\Sigma}(\mathbf{F}, \Sigma)\right] - \lambda, \quad (6)$$

Where $B$ is the number of minibatches in the dataset, $\lambda^0$ is the parameter for the prior distribution $p(\mathbf{F}, \Sigma)$, and $t_{\mathbf{F},\Sigma}(\mathbf{F}, \Sigma)$ is the sufficient statistic function for $p(\mathbf{F}, \Sigma)$. Thus, we can use this equation to compute the natural gradient update for $\lambda$, whereas for $\gamma$ and $\phi$ we use stochastic gradient updates on Monte Carlo estimates of the ELBO, specifically using the Adam optimization scheme (Kingma & Ba, 2015). This leads to two simultaneous optimizations for the PGM parameters and the neural network parameters, and their learning rates are treated as separate hyperparameters. We have found $10^{-3}$ and $10^{-4}$ to be generally suitable for the natural gradient and Adam updates, respectively.

---

**Algorithm 1** SOLAR

1: Hyperparameters: # iterations $K$, # trajectories $N$, model training buffer size $B$
2: Initialize policy $\pi_\theta^{(0)}$, model $\mathcal{M}^{(0)}$
3: **for** iteration $k \in \{1, \dots, K\}$ **do**
4:      Collect rollouts from the real world $\mathcal{D}^{(k)} = \{(\mathbf{s}_0^{(i)}, \mathbf{a}_0^{(i)}, \dots, \mathbf{s}_T^{(i)})\}_{i=1}^N$
5:      $\mathcal{M}^{(k)} \leftarrow \text{MODELUPDATE}(\mathcal{M}^{(k-1)}, \{\mathcal{D}^{(i)}\}_{i=k-B}^k)$ (section 3)
6:      $\tilde{\pi}_\theta^{(k-1)} \leftarrow \text{LINEARIZEPOLICY}(\mathcal{D}^{(k)}, \mathcal{M}^{(k)})$ (Appendix C)
7:      $\{\mathbf{F}_t^{(k)}, \Sigma_t^{(k)}\}_t \leftarrow \text{INFERDYNAMICS}(\mathcal{D}^{(k)}, \mathcal{M}^{(k)})$ (subsection 4.1)
8:      $\pi_\theta^{(k)} \leftarrow \text{POLICYUPDATE}(\tilde{\pi}_\theta^{(k-1)}, \{\mathbf{F}_t^{(k)}, \Sigma_t^{(k)}\}_t, \mathcal{M}^{(k)})$ (subsection 4.2)
9: **end for**

---

Figure 2 details the graphical model presented in Equation 2-Equation 4 along with the variational family described above. Since we are interested in control and RL, there is the added notion of observed costs from the environment, and there are many ways we could model these additional observations. A natural choice is to model costs as a quadratic function of the latent state and action, such that we arrive at the LQS presented in Equation 1 except in the learned latent space. Specifically, given trajectories of the form $[\mathbf{s}_0, \mathbf{a}_0, c_0, \dots, \mathbf{s}_{T-1}, \mathbf{a}_{T-1}, c_{T-1}, \mathbf{s}_T]$, we first embed the observations $\{\mathbf{s}_t\}$ using the mean of our recognition model $\mu(e_\phi(\mathbf{s}))$ to obtain a set of latent states $\{\mathbf{z}_t\}$. We then model our cost samples as $c_t = \frac{1}{2}\mathbf{z}_t^\top \mathbf{L}\mathbf{L}^\top \mathbf{z}_t + \mathbf{c}^\top \mathbf{z}_t + \alpha\|\mathbf{a}_t\|_2^2 + b$, where we assume that the action-dependent part of the cost is known and we learn $\mathbf{L}$, $\mathbf{c}$, and $b$ by minimizing the mean-squared error of the observed costs with stochastic gradient descent. $\mathbf{L}$ is a lower-triangular matrix with strictly positive diagonal entries, and thus by constructing our cost matrix as $\mathbf{C} = \mathbf{L}\mathbf{L}^\top$ we guarantee that the learned cost matrix is positive definite, which improves the conditioning of the policy update.

## 4 POLICY LEARNING IN THE LATENT SPACE

While we could use a variety of model-based policy learning methods in the learned latent space, the ability to infer local time-varying linear dynamics lends itself naturally to the particular analytic local solution to the policy described in subsection 2.1. This approach yields a policy that is TVLG in the latent space, which in general corresponds to a class of nonlinear policies in the original space formed by the composition of the nonlinear neural network embedding and the TVLG policy.

As discussed in the following sections, we can use the PGM in the previous section to formulate local model fitting as probabilistic inference, in order to obtain a dynamics estimate that we can then use to improve the policy. Note that this use of the model is quite different from how dynamics models are typically used in standard model-based RL algorithms: instead of using the model to predict into the future, we only use the model to infer local linear dynamics conditioned on real-world trajectory samples. While local models are not burdened by forward prediction compared to global forward models, the simplicity of linear local models prevents accurate modeling of complex systems, and our method mitigates this through a latent representation that is optimized for local linear model fitting.

Our overall algorithm, SOLAR, is presented in algorithm 1. At every iteration, we collect $N$ rollouts from the real world (line 4). Then, we update our model using data from the last $B$ iterations (line 5), we linearize our policy given the updated model (line 6, see Appendix C for details), we perform inference within our model to get the dynamics estimates (line 7), and we update our policy using the rollouts from our current iteration and our updated model (line 8). The following subsections detail the modules of our method that are involved in policy learning and improvement.

### 4.1 DYNAMICS INFERENCE UNDER THE MODEL

To obtain a TVLG dynamics model, we could directly use linear regression to fit $\mathbf{F}_t$ and $\Sigma_t$ to the observed latent trajectories $\tau = [\mathbf{z}_0, \mathbf{a}_0, \dots, \mathbf{z}_{T-1}, \mathbf{a}_{T-1}, \mathbf{z}_T]$. However, this may be poorly conditioned in the low-data regime. Instead, we can perform inference within our model to obtain dynamics estimates for policy improvement. As described in section 3, our model provides us with variational approximations to the posterior over dynamics models, i.e., $\{q(\mathbf{F}_t, \Sigma_t)\}_{t=0}^{T-1}$, which are MNIW. We can use these as a prior and condition on the data to obtain new variational posteriors

$\{q(\mathbf{F}_t, \Sigma_t | \{\tau\}_{i=0}^N)\}_{t=0}^{T-1}$, which are also MNIW. Writing the parameters of these posteriors – for which the closed form solutions are given in Appendix B– as $\{\Psi_t, \boldsymbol{M}_{0t}, \boldsymbol{V}_t, \nu_t\}_t$, we compute a maximum a posteriori estimate of the dynamics parameters at time step $t$ as: $\mathbf{F}_t = \boldsymbol{M}_{0t}$, $\Sigma_t = \frac{\Psi_t}{\nu_t}$. This inference procedure corresponds to Bayesian linear regression and can be interpreted as resolving the uncertainty in the global dynamics model conditioned on a real-world rollout. In essence, $\{q(\mathbf{F}_t, \Sigma_t)\}_{t=0}^{T-1}$ captures uncertainty over the latent system dynamics by acting as a global model over all observed data, but in order to accurately model the system within the local region around the current policy, we condition on trajectories collected from the policy in order to resolve the uncertainty and obtain dynamics estimates $\{\mathbf{F}_t, \Sigma_t\}_{t=0}^{T-1}$ that allow us to improve the policy.

## 4.2 POLICY UPDATE

As described in subsection 2.1, once we have our TVLG dynamics estimates $\{\mathbf{F}_t, \Sigma_t\}_t$ and quadratic cost fit $\mathbf{C}, \mathbf{c}$, we can use dynamic programming on the Q and value functions to compute the optimal policy in closed form. However, doing so is typically undesirable as the resulting policy will overfit to the model and likely will not perform well in the real environment. Since our modeling assumption is not that our model will be globally valid, but rather that our model will be valid close to the data distribution of the previous policy, we utilize a constrained policy update such that our new policy does not drastically change the induced trajectory distribution. Specifically, similar to prior work, we impose a KL-divergence constraint on the policy update such that the shift in the induced trajectory distributions before and after the update, which we denote as $\bar{p}(\tau)$ and $p(\tau)$, respectively, is bounded by a step size $\epsilon$ (Levine & Abbeel, 2014). This leads to a constrained optimization of the form $\max_\theta \hat{\eta}(\theta)$ s.t. $D_{\text{KL}}(p(\tau) \| \bar{p}(\tau)) \leq \epsilon$. As shown in Levine & Abbeel (2014), this constrained optimization can be solved by augmenting the cost function to penalize the deviation from the previous policy $\pi_{\bar{\theta}}$, i.e., $\tilde{C}(\mathbf{z}_t, \mathbf{a}_t) = \frac{1}{\lambda} C(\mathbf{z}_t, \mathbf{a}_t) - \log \pi_{\bar{\theta}}(\mathbf{a}_t | \mathbf{z}_t)$. Note that this augmented cost function is still quadratic, since the policy is TVLG, and thus we can still compute the optimal policy under this cost function in closed form using the procedure described in subsection 2.1. $\lambda$ is a dual variable that trades off between optimizing the cost function and staying close in distribution to the previous policy, and the weight of this term can be determined through a dual gradient descent procedure. Combined with the model learning from section 3, we arrive at the SOLAR algorithm.

## 5 RELATED WORK

Model-based RL methods have achieved significant efficiency benefits compared to model-free RL methods (Chebotar et al., 2017; Nagabandi et al., 2018; Deisenroth et al., 2014). Many of these prior methods learn global models of the system that are then used for planning, generating synthetic experience, or policy search (Atkeson & Schaal, 1997; Peters et al., 2010). These methods require an accurate and reliable model and will typically suffer from modeling bias, hence these models are still limited to short horizon prediction in more complex domains (Mishra et al., 2017; Nagabandi et al., 2018; Gu et al., 2016; V.Feinberg et al., 2018). Another class of model-based methods rely only on local system models to compute the gradient for a policy update (An et al., 1988; Kolter & Ng, 2005; Heess et al., 2015; Levine & Abbeel, 2014; Bansal et al., 2017). These methods do not use models for long-term forward prediction, allowing for the use of simple models that enable policy improvement (Montgomery et al., 2017; Levine et al., 2016). As we show in section 6, modeling bias for prior methods can be severely limiting in systems with complex observations such as images, whereas we are able to learn representations that mitigate the effects of modeling bias.

Utilizing representation learning within model-based RL has been studied in a number of previous works (Lesort et al., 2018), including using embeddings for state aggregation (Singh et al., 1994), dimensionality reduction (Nouri & Littman, 2010), self-organizing maps (Smith, 2002), value prediction (Oh et al., 2017), and deep auto-encoders (Lange & Riedmiller, 2010; Finn et al., 2016; Watter et al., 2015; Higgins et al., 2017). Within these works, deep spatial auto-encoders (DSAE) (Finn et al., 2016) and embed to control (E2C) (Watter et al., 2015; Banijamali et al., 2017) are the most closely related to our work in that they consider local model methods combined with representation learning. The key difference in our work is that, rather than using a learning objective for reconstruction and forward prediction, we formulate a Bayesian latent variable model such that inference corresponds to fitting local models within the learned representation. As such, our objective enables local model methods by directly encouraging learning representations where fitting local models accurately

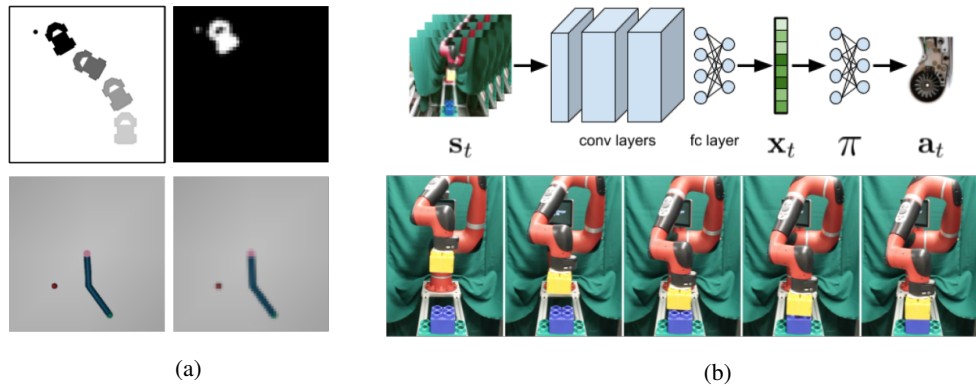

(a)  (b)

Figure 3: (a) Top: Visualizing a trajectory in the car navigation environment, with the target denoted by the black dot, and the corresponding image observation. Bottom: An illustration of the 2-DoF arm environment, with the target denoted by the red dot, and the corresponding image observation. Note that we use sliding windows of past observations when learning both tasks. (b) Top: Illustration of the architecture we use for learning Lego block stacking. Bottom: Example trajectory from our learned policy stacking the yellow Lego block on top of the blue block.

explains the observed data. We also do not assume a known cost function, goal state, or access to the underlying system state as in DSAE and E2C, thus SOLAR is applicable even when the underlying states and cost function are unknown.[1] We find that our approach tends to produce better results on a number of complex image-based tasks, as we discuss in the next section.

## 6 EXPERIMENTS

We aim to answer the following questions through our experiments: (1) How does SOLAR compare to state-of-the-art model-free and model-based RL algorithms? (2) How do local and global model methods compare when operating in our learned representations? (3) How much benefit do we derive from our particular representation learning method? To answer (1), we compare SOLAR to trust region policy optimization (TRPO) (Schulman et al., 2015) and proximal policy optimization (PPO) (Schulman et al., 2017), two state-of-the-art model-free methods, and LQR with fitted linear models (LQR-FLM) (Levine & Abbeel, 2014), a state-of-the-art model-based method. To answer (2), we test an ablation of our method where we learn a neural network dynamics model with which we perform model-predictive control (MPC) in the latent space. We refer to this as the "global model ablation". To answer (3), we replace our LDS SVAE model with a variational auto-encoder (VAE) (Kingma & Welling, 2014; Rezende et al., 2014) and with the robust locally-linear controllable embedding (RCE) model (Banijamali et al., 2017), an improved version of the E2C model (Watter et al., 2015). We refer to these as the "VAE ablation" and "E2C-like ablation", respectively. We additionally compare to a pixel space model similar to Finn & Levine (2017) that utilizes no representation learning and instead learns both a dynamics and cost model on images in order to run MPC in pixel space. Videos of the learned policies are available on the project website.[2]

### 6.1 EXPERIMENTAL TASKS

We set up simulated image-based robotic domains for a 2-dimensional navigation task, a nonholonomic car, and a 2-DoF arm, as shown in Figure 3a. We also learn a block stacking task directly from camera images on a real Sawyer robotic arm, as shown in Figure 3b. Details regarding experimental setup and training hyperparameters are provided in Appendix D.

---

[1]In principle, these methods can be extended to unknown underlying states and cost functions, though the authors do not experiment with this and it is unclear how well these approaches would generalize.

[2]https://sites.google.com/view/iclr19solar

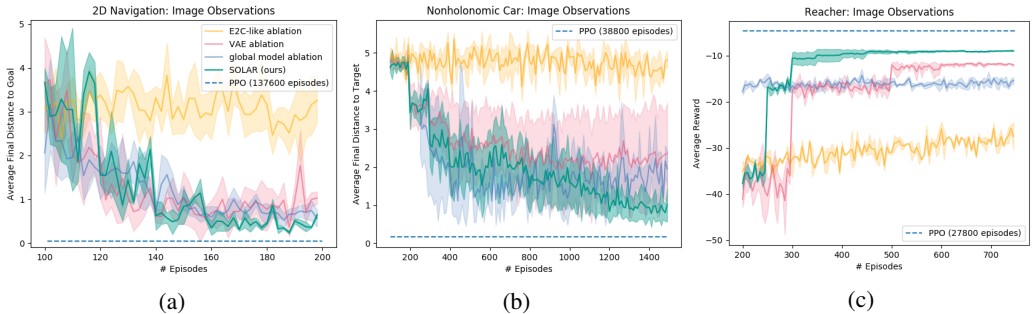

Figure 4: (a) Our method, the VAE ablation, and the global model ablation consistently solve 2D navigation from images, whereas LQR-FLM and the E2C-like ablation are unable to make progress. The final performance of PPO is plotted as the dashed line, though PPO requires 1000 times more samples than our method to reach this performance. (b) On the car from images, both our method and the global model ablation are able to reach the goal, however, we encode prior information into the global model ablation by biasing the control to select positive actions. The VAE ablation is less consistent across random seeds, and the E2C-like ablation once again is unsuccessful at the task. PPO requires over 25 times more episodes to learn a successful policy. (c) For reacher from images, we perform worse than PPO but need about 40 times fewer episodes to learn, whereas the ablations performs noticeably worse. Here we plot reward, so higher is better.

**2D navigation.** We consider a 2-dimensional navigation task similar to Watter et al. (2015); Banijamali et al. (2017) except we move the goal every episode rather than fixing it to the bottom right. Observations consist of two 32-by-32 images indicating the positions of the agent and goal.

**Nonholonomic car.** The nonholonomic car starts in the bottom right of the 2-dimensional space and controls its acceleration and steering velocity in order to reach the target in the top left. We use a sliding window of four 64-by-64 images as the observation to capture velocity information.

**Reacher.** We experiment with the reacher environment from OpenAI Gym (Brockman et al., 2016), where a 2-DoF arm has to reach a target denoted by a red dot, which we specify to be in the bottom left. For observations, we directly use 64-by-64-by-3 images of the rendered environment, which provides a top-down view of the reacher and target, and we use a sliding window of four images.

**Sawyer Lego block stacking.** To demonstrate a challenging task in the real world, we use our method to learn Lego block stacking with a real 7-DoF Sawyer robotic arm, as depicted in Figure 3b. The observations used are raw 84-by-84-by-3 images from a camera pointed at the robot, and the controller only receives images as the observation, without joint angles or other information.

## 6.2 SIMULATION RESULTS

Figure 4 details our results on the simulated image-based experimental domains, where each method is tested on three random seeds and the mean and standard deviation of the performance is reported. For the 2D navigation and car tasks from images, we plot the average final distance to the goal as a function of the number of episodes, so lower is better.[3] On the reacher task, we plot the reward function as defined by Gym since this is the standard metric used to evaluate performance on this task, and as shown by the videos on our project website, achieving high Gym reward correlates strongly with solving the task in terms of distance to the goal.

On 2D navigation from images, our method, the VAE ablation, and the global model ablation are all able to learn very quickly, converging to high-performing policies within 200 episodes. LQR-FLM struggles to learn the task, likely because the images are too complex for local linear model fitting, and makes no progress at all. In fact, LQR-FLM fails to learn on all of the simulated tasks, and we note that this precludes the guided policy search (GPS) method from solving these tasks (Levine et al., 2016), as GPS uses LQR-FLM as a subroutine. For the sake of clarity in the plots, we omit the

---

[3]We plot the ground truth distance for the 2D navigation, car, and Sawyer block stacking tasks for evaluation purposes only, and this information is not available to the learning algorithms.

LQR-FLM results, which are qualitatively similar to the E2C-like ablation results. PPO eventually learns a successful policy, as indicated by the dashed line depicting this method's final performance, but this requires roughly three orders of magnitude more samples than our method. We present log-scale plots that illustrate the full learning progress of model-free methods in Appendix E.

Despite using code directly from the authors of RCE, we were unable to get the E2C-like ablation to learn a good model for this task, and thus the learned policy does not improve over the initial policy. In fact, we were unable to learn successful policies for any of the simulated tasks, though in Appendix E, we demonstrate that this ablation can learn a more successful policy on the 2D navigation domain used by Watter et al. (2015); Banijamali et al. (2017), where the target is fixed to the bottom right. This highlights the difficulty of the tasks we consider.

On the image-based car, our method is able to learn a good policy with about 1500 episodes of experience. The global model ablation is competitive with our method, however, we obtained this result by biasing the mean of the MPC random action selection to be positive, effectively encoding prior information that the car should move forward. We also noticed that, even with more data, the variance of the MPC performance remained higher than the policy learned by our method. These observations indicate that forward prediction using the learned global models may be inaccurate, leading to inconsistent control performance. In contrast, our method does not heavily rely on an accurate model and can achieve consistently good behavior on this task. The VAE ablation is able to solve this task for some random seeds, however this method's performance is less consistent compared to our method. PPO eventually learns a successful policy for this task that performs better than our method, however it uses over 25 times more data than our method.

Finally, on the image-based reacher task, our method achieves worse final policy performance than PPO, though we do so with about 40 times fewer episodes, i.e., we use under 700 episodes whereas PPO uses about 30000. This gain in data efficiency compared to model-free methods is typical of model-based methods, however, SOLAR is able to handle this domain directly from raw image observations, which is challenging for other model-based methods. The VAE ablation also makes progress toward the goal, however, the performance is noticeably worse compared to our method. The global model ablation makes very little improvement over its initial behavior, which is better than the other methods as it learns both a dynamics and cost model from the pretraining data and uses these models right away for planning. This performance drop compared to the previous tasks indicates the difficulty in forward prediction for this domain, coupled with the failure of short-horizon control for this task as greedily minimizing distance to the goal often simply leads to collapsing the arm. As it is also less intuitive to encode prior information into this task compared to biasing the actions in the car domain to drive forward, we could not get this ablation to succeed on this task.

## 6.3 REAL ROBOT RESULTS

Figure 5 details performance on the Lego block stacking tasks in terms of the average final distance in centimeters to the goal, where we test on three random seeds and report the mean and standard deviation of the performance. We define the goal position of the end effector such that reaching the goal leads to successful stacking of the block. Not only is our method able to solve this task directly from raw, high-dimensional camera images within 200 episodes, corresponding to about half an hour of interaction time, our method is also successful at handling the complex, contact-rich dynamics of block stacking. As seen in the video on our project website, our method learns a policy that can react to slightly different contacts, due to the bottom block shifting between episodes, and is ultimately successful in stacking the block in most episodes.[4]

We compare to the VAE and global model ablations, as these proved to be the most successful and data efficient baselines in simulation. These ablations are competitive

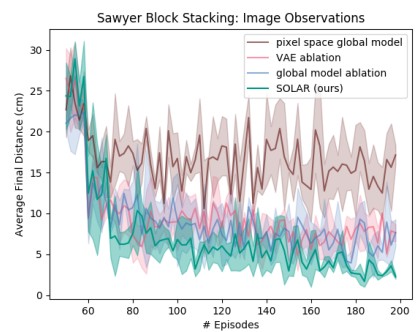

Figure 5: Performance on the real-world Sawyer block stacking task. Our method learns to successfully stack the block in about half an hour of interaction time. The VAE and global model ablations are also competitive on this task, while the pixel space model performs worse.

---

[4]https://sites.google.com/view/iclr19solar

with our method for this real world task, though our method still achieves a better final policy that is able to more consistently stack the block. The pixel space model is significantly worse than the other methods that learn a latent representation, and given prior work on pixel space global models (Finn & Levine, 2017), we suspect that this method would need more data in order to learn this task.

# 7 DISCUSSION AND FUTURE WORK

We presented SOLAR, a model-based RL algorithm that is capable of learning policies in a data-efficient manner directly from raw high-dimensional observations. The key insights in SOLAR involve learning latent representations where simple models are more accurate and utilizing PGM structure to infer dynamics from data conditioned on entire real-world trajectories. Our experimental results demonstrate that SOLAR is competitive in sample efficiency, while exhibiting superior final policy performance, compared to other model-based methods. Furthermore, SOLAR is significantly more data-efficient compared to state-of-the-art model-free RL methods.

There are several interesting directions for future work. First, the ability to learn representations lends itself naturally to multi-task and transfer settings, where new tasks could potentially be learned much more quickly by starting from a latent embedding that has been learned from previous tasks. We can also in principle share dynamics models, where the PGM we learn from solving previous tasks can be used as a global prior when inferring local dynamics fits for a new task. Second, our model is designed for and tested on continuous action domains as we focus on robotic applications. Extending our model to discrete actions would necessitate some type of continuous relaxation or learned action representation, and we believe that this is another interesting direction for future work.

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

## A    POLICY LEARNING DETAILS

Given a TVLG dynamics model and quadratic cost approximation, we can approximate our Q and value functions to second order with the following dynamic programming updates, which proceed from the last time step $t = T$ to the first step $t = 1$:

$$Q_{\mathbf{s},t} = c_{\mathbf{s},t} + \mathbf{F}_{\mathbf{s},t}^{\top} V_{\mathbf{s},t+1}, \;\; Q_{\mathbf{ss},t} = c_{\mathbf{ss},t} + \mathbf{F}_{\mathbf{s},t}^{\top} V_{\mathbf{ss},t+1} \mathbf{F}_{\mathbf{s},t},$$

$$Q_{\mathbf{a},t} = c_{\mathbf{a},t} + \mathbf{F}_{\mathbf{a},t}^{\top} V_{\mathbf{s},t+1}, \;\; Q_{\mathbf{aa},t} = c_{\mathbf{aa},t} + \mathbf{F}_{\mathbf{a},t}^{\top} V_{\mathbf{ss},t+1} \mathbf{F}_{\mathbf{a},t},$$

$$Q_{\mathbf{sa},t} = c_{\mathbf{sa},t} + \mathbf{F}_{\mathbf{s},t}^{\top} V_{\mathbf{ss},t+1} \mathbf{F}_{\mathbf{a},t},$$

$$V_{\mathbf{s},t} = Q_{\mathbf{s},t} - Q_{\mathbf{sa},t} Q_{\mathbf{aa},t}^{-1} Q_{\mathbf{a},t},$$

$$V_{\mathbf{ss},t} = Q_{\mathbf{ss},t} - Q_{\mathbf{sa},t} Q_{\mathbf{aa},t}^{-1} Q_{\mathbf{as},t}.$$

It can be shown (e.g., by Tassa et al. (2012)) that the action $\mathbf{a}_t$ that minimizes the second-order approximation of the Q-function at every time step $t$ is given by

$$\mathbf{a}_t = -Q_{\mathbf{aa},t}^{-1} Q_{\mathbf{as},t} \mathbf{s}_t - Q_{\mathbf{aa},t}^{-1} Q_{\mathbf{a},t}.$$

This action is a linear function of the state $\mathbf{s}_t$, thus we can construct an optimal linear policy by setting $\mathbf{K}_t = -Q_{\mathbf{aa},t}^{-1} Q_{\mathbf{as},t}$ and $\mathbf{k}_t = -Q_{\mathbf{aa},t}^{-1} Q_{\mathbf{a},t}$. We can also show that the maximum-entropy policy that minimizes the approximate Q-function is given by

$$\pi(\mathbf{a}_t | \mathbf{s}_t) = \mathcal{N}(\mathbf{K}_t \mathbf{s}_t + \mathbf{k}_t, Q_{\mathbf{aa},t}).$$

Furthermore, as in Levine & Abbeel (2014), we can impose a constraint on the total KL-divergence between the old and new trajectory distributions induced by the policies through an augmented cost function $\bar{c}(\mathbf{s}_t, \mathbf{a}_t) = \frac{1}{\lambda} c(\mathbf{s}_t, \mathbf{a}_t) - \log \pi^{(i-1)}(\mathbf{a}_t | \mathbf{s}_t)$, where solving for $\lambda$ via dual gradient descent can yield an exact solution to a KL-constrained LQR problem.

## B    DYNAMICS INFERENCE

Here we provide the closed form parameter computations for the posteriors of our dynamics given observed trajectories, as described in Section 4.1 of the main paper. Given variational factors from our model of the form

$$q(\mathbf{F}_t, \Sigma_t) = \mathcal{MNIW}(\Psi_t', \nu_t', \mathbf{M}_{0t}', \mathbf{V}_t') \text{ for } t \in [0, \ldots, T-1],$$

We can condition on observed trajectories $\tau$ to obtain new variational posteriors $\{q(\mathbf{F}_t, \Sigma_t | \{\tau\}_{i=0}^N)\}_{t=0}^{T-1}$. These posteriors are also MNIW, and the parameters of these posteriors can be computed in closed form as

$$\Psi_t = \Psi_t' + \mathbf{M}_{0t}' \mathbf{V}_t'^{-1} \mathbf{M}_{0t}'^{\top} + \sum_{i=1}^{N} \mathbf{z}_{t+1}^{(i)} \mathbf{z}_{t+1}^{(i)\top} - \mathbf{M}_{0t} \mathbf{V}_t^{-1} \mathbf{M}_{0t}^{\top}, \qquad \kappa_t = \kappa_t + N,$$

$$\mathbf{M}_{0t} = \left( \mathbf{M}_{0t}' \mathbf{V}_t'^{-1} + \sum_{i=1}^{N} \mathbf{z}_{t+1}^{(i)} \begin{bmatrix} \mathbf{z}_t^{(i)} \\ \mathbf{a}_t^{(i)} \end{bmatrix}^{\top} \right) \mathbf{V}_t, \qquad \mathbf{V}_t = \left( \mathbf{V}_t'^{-1} + \sum_{i=1}^{N} \begin{bmatrix} \mathbf{z}_t^{(i)} \\ \mathbf{a}_t^{(i)} \end{bmatrix} \begin{bmatrix} \mathbf{z}_t^{(i)} \\ \mathbf{a}_t^{(i)} \end{bmatrix}^{\top} \right)^{-1}.$$

Then, a maximum a posteriori estimate gives us the TVLG dynamics parameters as described in the main paper.

## C    POLICY LINEARIZATION

The policy update described in Section 4.2 of the main paper requires us to compute the KL-divergence between the trajectory distributions before and after the policy update, denoted as $\bar{p}(\tau)$ and $p(\tau)$, respectively. We compute $p(\tau) = \hat{\rho}(\mathbf{z}_0) \prod_{t=0}^{T-1} \pi_\theta(\mathbf{a}_t | \mathbf{z}_t) \hat{p}(\mathbf{z}_{t+1} | \mathbf{z}_t, \mathbf{a}_t)$, and analogously for $\bar{p}(\tau)$ with the previous policy, and we are able to compute these analytically because the policies and dynamics model are TVLG, thus the induced trajectory distributions are also Gaussian. However, this operates under the assumption that $\mathbf{z}$ is fixed, which does not hold since the model update changes

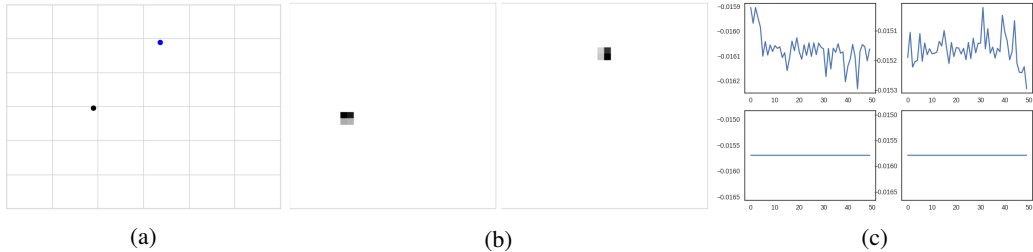

Figure 6: (a) An illustration of the 2D navigation task, with the agent depicted as the black dot and the target depicted as the blue dot. (b) We use as observations two 32-by-32 images stacked on top of each other, where the first observation indicates the position of the agent the second observation indicates the position of the target. (c) Visualization of the 4-dimensional latent space for an example random trajectory of the 2D-navigation task. Note that the range of values in the latent space is very narrow, and the bottom two dimensions seemingly capture information about the target which does not move.

the latent representation. Since our overall policy is a combination of the model embedding, given by $e_\phi(\mathbf{s})$, and the TVLG policy $\pi_\theta(\mathbf{a}_t|\mathbf{z}_t)$, training $e_\phi(\mathbf{s})$ will change the behavior of the policy even if $\pi_\theta(\mathbf{a}_t|\mathbf{z}_t)$ stays fixed. In some cases, this may lead to a policy with worse performance, and constraining against this policy for the policy update may lead to poor results. In fact, what we want to do is to account for the model update by changing $\pi_\theta(\mathbf{a}_t|\mathbf{z}_t)$ accordingly, so that the overall policy does not change in its distribution. Thus, using $(\mathbf{s}_t, \mathbf{a}_t)$ pairs from the previous data collection phase, we embed $\mathbf{z}_t = \mu(e_\phi(\mathbf{s}_t))$ with our updated model and use linear regression to find the TVLG policy $\tilde{\pi}_\theta(\mathbf{a}_t|\mathbf{z}_t)$ that best explains the data collected from the policy This is line 6 of the SOLAR algorithm presented in the main paper, and after this, we can perform the policy update constrained against the trajectory distribution induced by $\tilde{\pi}_\theta(\mathbf{a}_t|\mathbf{z}_t)$.

## D    EXPERIMENT SETUP

**Image-based 2D navigation.** Our recognition model architecture for the 2D navigation domain consists of two convolution layers with 2-by-2 filters and 32 channels each, with no pooling layers and ReLU non-linearities, followed by another convolution with 2-by-2 filters and 2 channels. The output of the last convolution layer is fed into a spatial softmax layer (Finn et al., 2016), which then outputs a Gaussian distribution with a fixed diagonal covariance of $10^{-4}$ for the latent distribution. Our observation model consists of two fully-connected (FC) hidden layers with 256 ReLU activations, and the last layer outputs a categorical distribution over pixels. We initially collect 200 episodes which we use to train our model, and for every subsequent iteration we collect 20 episodes to fine tune our model. The cost function we use is the sum of the $L^2$-norm squared of the distance to the target and the commanded action, with weights of 1 and 0.001, respectively.

**Image-based nonholonomic car.** The image-based car domain consists of 64-by-64 image observations. We include a window of the 3 previous 64-by-64 images in our observation to preserve velocity information. Our recognition model is a convolutional neural network that operates on each image in the sliding window independently. Its architecture is four convolutional layers with 4-by-4 filters with 4 channels each, and the first two convolution layers are followed by a ReLU non-linearity. The output of the last convolutional layer is fed into three FC ReLU layers of width 2048, 512, and 128, respectively. Our final layer outputs a Gaussian distribution with dimension 8. This leads to a final latent dimension of 32. Our observation model consists of four FC ReLU layers of width 256, 512, 1024, and 2048, respectively, followed by a Bernoulli distribution layer that models the image. Like the recognition model, the observation model only operates on each section of the latent representation corresponding to the image window independently. For this domain, we collect 100 episodes initially to train our model, and we collect 100 episodes per iteration after this. The cost function we use is the sum of the $L^2$-norm squared of the distance from the center of the car to the target and the commanded action, with weights of 1 and 0.001, respectively.

**Reacher.** The reacher domain consists of 64-by-64-by-3 image observations. Similar to the car, we include a window of the 3 previous 64-by-64-by-3 images in our observation. Our recognition

model is a convolutional neural network that again operates on each image in the sliding window independently. Its architecture is three convolutional layers with 2-by-2 filters with 64, 32 and 16 channels respectively. Each layer has a ReLU non-linearity followed by a 2-by-2 max-pooling. The output of the last convolutional layer is fed into an FC ReLU layer of width 200, followed by another FC ReLU layer of width 200. Our final layer outputs a Gaussian distribution with dimension 10, leading to a final latent dimension of 40. Our observation model consists of three FC ReLU layers of width 256, followed by a Bernoulli distribution layer and separately models each image in the sliding window. We collect 200 episodes initially to train our model, and we collect 100 episodes per iteration after this. The cost function we use is the sum of the $L^2$-norm of the distance from the fingertip to the target and the $L^2$-norm squared of the commanded action.

**Sawyer Lego block stacking.** The image-based Sawyer block-stacking domain consists of 84-by-84-by-3 image observations. The policy outputs velocities on the end effector in order to control the robot. Our recognition model is a convolutional neural network with the following architecture: a 5-by-5 filter convolutional layer with 16 channels followed by two convolutional layers using 5-by-5 filters with 32 channels each. The first two convolutional layers are followed by ReLU activations and the last by a FC ReLU layer of width 256 leading to a 16 dimensional Gaussian distribution layer. Our observation model consists of a FC ReLU layer of width 128 feeding into three deconvolutional layers, the first with 5-by-5 filters with 32 channels and the last two of 6-by-6 filters with 16 and 3 channels respectively. These are followed by a final Bernoulli distribution layer. For this domain, we collect 50 episodes initially to train our model, 20 episodes per iteration for the first 5 iterations, then 10 episodes per iteration for the remainder. The cost function is the sum of the $L^1$-norm of a weighted displacement vector between the end-effector and the target in 3D-space (weighted 1, 2, 1 for $x$, $y$, $z$), the $L^2$-norm in the same space, and the angle of rotation required to reach a valid wrist orientation, with weights of 1, .1, and .15, respectively.

## E  ADDITIONAL EXPERIMENTS

### E.1  E2C-LIKE ABLATION ON SIMPLIFIED 2D NAVIGATION

As mentioned in Section 6, our E2C-like ablation was unable to make progress for the 2D navigation task, though we were able to get more successful results by fixing the position of the goal to the bottom right as is done in the image-based 2D navigation task considered in E2C (Watter et al., 2015) and RCE (Banijamali et al., 2017). Figure 7 details this experiment, which we ran for three random seeds and report the mean and standard deviation of the average final distance to the goal as a function of the number of training episodes. It is clear that the policy is improving, and two of the seeds are able to make substantial progress, though the final seed is less successful and significantly worsens the average performance of the method. This indicates that the latent representation learned through RCE is less suitable for local model fitting, as accurate local model fitting is not explicitly encouraged by their representation learning objective.

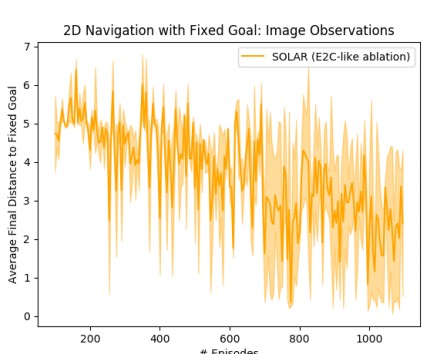

Figure 7: On 2D navigation with the goal fixed to the bottom right, our E2C-like ablation is able to make progress toward the goal.

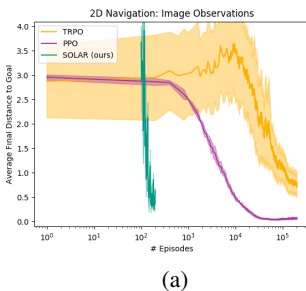 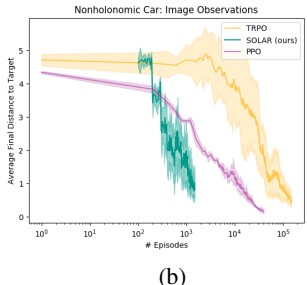 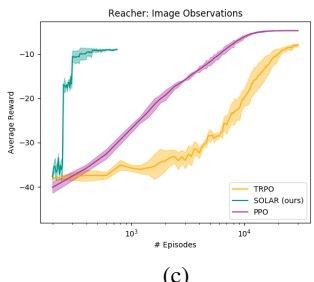

(a)             (b)             (c)

Figure 9: (a) Comparison of our method to PPO on the 2D navigation task presented in the paper. Our method uses roughly three orders of magnitude fewer samples to solve the task compared to PPO. (b) On the car from images task, our method achieves slightly worse performance than PPO though with about 25 times fewer samples. (c) Comparison of our method to TRPO and PPO for the reacher task. Our method achieves slightly worse final performance but uses about 40 times fewer samples than these methods.

## E.2 MODEL-BASED COMPARISONS ON STATE-BASED NONHOLONOMIC CAR

To provide a point of comparison to model-based RL methods, we consider the car domain where the underlying state is observed. The states for the car domain include the position of the center of mass, orientation, forward and angular velocity of the car, and the position of the target, making for a 9-dimensional system. Since this observation is already quite simple, we use a single linear layer for our recognition and observation models that output Gaussian distributions, and we use the same dimensionality for our latent representation as the state.

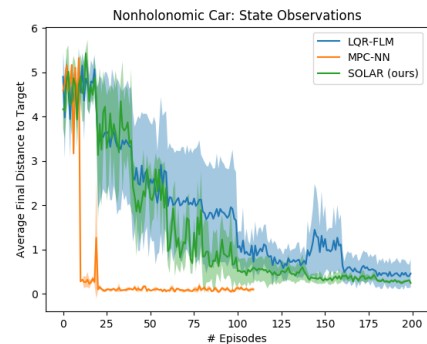

Figure 8: On the car from states, our method is competitive with LQR-FLM, demonstrating that we maintain the sample efficiency of model-based methods for simple tasks.

We plot the performances of our method, LQR-FLM (Levine & Abbeel, 2014), and Nagabandi et al. (2018), which we refer to as model-predictive control with neural networks (MPC-NN), again based on the average final distance to the target, in Figure 8. In this setting, our method is competitive with LQR-FLM, learning a policy with similar performance in 200 episodes. MPC-NN performs the best for this task, learning a policy that consistently reaches the target in just 20 episodes, though it is given the true cost function whereas our method and LQR-FLM are not. For this simple setup where modeling bias is not an issue, we expect model-based methods to perform very well and learn efficiently. However, when we make the problem more challenging by using image observations, model-based methods will fail quickly: LQR-FLM is unable to fit complex pixel transitions using local linear models, as shown through the 2D navigation experiment, and MPC-NN has never been used with images, as forward video prediction and defining a cost function on images are both very difficult. We extend MPC-NN to the image-based task, and we term this the "global model ablation" of our method – as shown in the paper, this approach is able to make progress toward the goal, though our method is still significantly better at solving this difficult task.

## E.3 FULL PERFORMANCE OF TRPO ON 2D NAVIGATION AND REACHER

In Figure 9 we include the plots for the simulated tasks comparing SOLAR, PPO, and TRPO. Note that the x-axis is on a log scale, i.e., though our method is sometimes worse in final policy performance to PPO and TRPO, we do so with one to three orders of magnitude fewer samples. This demonstrates our method's sample efficiency compared to model-free methods, while being able to solve complex image-based domains that are difficult for model-based methods.

