# OpenReview forum: "SOLAR: Deep Structured Representations for Model-Based Reinforcement Learning"
_ICLR.cc/2019/Conference_

### Official Review · AnonReviewer3 · 2018-11-02
**Nice work but methodology is not clear enough in some parts. It can also benefit from a broader experimental evaluation.**

**Rating:** 5
**Confidence:** 4

**Review:**

In this work the authors propose an end to end approach for model based reinforcement learning from images, where the main building blocks are locally-linear dynamical systems and variational auto-encoders (VAE). Specifically, it is assumed that the input features (i.e., the images) are generated from a low dimensional latent representation mapped through parametric random functions; the latter are modeled via neural networks. A recognition model based on convolutional neural networks operates on the reverse way and is responsible for projecting the input features to the latent space, in order to proceed with the reinforcement learning task. The variational framework is employed in order to jointly learn the VAE and the linear dynamics on the latent state. As a final step, once the model is fitted a linear quadratic system (LQS) is solved in order to learn the cost function and the optimal policy.

* The paper is well motivated and tries to solve an interesting problem, that of data-efficient reinforcement learning. The experiments are well picked and demonstrate the advantages of the proposed approach towards solving the task, however, the method is only evaluated on few environments and compared against only a couple of other methods. I would expect a broader evaluation and/or comparison against more methods. Since the model is able to reach TRPO’s performance in much less steps it would be nice to see how it performs against PPO from [Schulman et al. 2017] (at least on the simulated environments). Also, would it make sense to compare against [Levine et al. 2016] that has been evaluated on similar tasks?

[Schulman et al 2017] “Proximal Policy Optimization Algorithms”.
[Levine et al. 2016] “End-to-End Training of Deep Visuomotor Policies”.

* Methodologically, the paper is sound. The model part (as the authors point out) is based on [Johnson et al. 2016] and is well explained. On the other hand, the policy part, and in particular the policy update in Section 4.2 has some issues regarding readability. There is a strong interplay between Section 4.2, Section 2.1 and Appendix D and the authors did not manage to nicely explain what exactly is happening during the update phase. In the beginning the reader has the impression that we are finding the optimal policy via the closed-form LQS. Later on we switch to constrained optimisation for the cost by accounting for the KL divergence between the policy on two episodes. Finally, in the appendix we are back to the original quadratic cost. The authors need to clarify all the above. Also, they need to explicitly mention why they opt for stochastic optimisation (is it because of minibatching?)

* To continue with the policy, in Section 4.2 the authors argue that although the optimal policy can be found in closed form this is not desirable because the policy will overfit the model and will not generalise well in the real environment. I disagree with this statement. If this happens it effectively means that the learned model or the assumption/learning of the linear dynamics is not right. The authors seem to also agree with this since they clearly state in the the experimental section that “... our method does not heavily rely on an accurate model...”. To my understanding, this means that we need to refine the modelling strategy and not learn a sub-optimal policy. I am really interested in the authors opinion on that.

* The above argument is also directly related to the recognition model and learning of the policy in the latent state (I completely agree with that). The recognition network, which in this case is a convolutional neural network, is used as an inference mechanism to project the observations to the latent space. We learn the (variational) parameters of the recognition model by optimising the likelihood’s lower bound. This means that we are “allowed” to overfit the variational parameters as long as the bound gets tighter. This can possibly result in degraded performance during the policy update. Furthermore, the variational distribution of the latent state, i.e., q(z_t | s_t) is assumed to be mean field across time (independent z’s), while clearly this is not the case in the posterior. You somehow mitigate that by augmenting the observed state (feeding consecutive frames to the network), but still this is not ideal. Finally, is there a reason why we only use the mean of the recognition model to fit the cost on the projected latent states? Why are we throwing away the uncertainty? Especially since you do not use an exact solver and follow a stochastic gradient.

* In the end of Section 2.1, the authors argue regarding the fact that the prior work assumes access to a compact low-dimensional representation which does not allow them to perform well on images. Reference is needed.

* In the related work the authors mention modelling bias as a downside of prior work. Can you please elaborate on that? Where does the bias come from and, more importantly, how does your approach overcome this issue?

* In the experiment and specifically in Figure 4 am I right in assuming that the distance to target is measured in actual pixels? Furthermore, why the relevant plot for the reacher task is depicting rewards instead of the distance to target. To me this suggests that the task is not solved. In general what I find very upsetting in the field are plots that only depict accumulated reward for a specific task. There are many situations where the agent learns a weird behaviour that happens to give good rewards (e.g., spinning around the cart-pole), and unfortunately such behaviours are not spotted on the reward plots.

Overall, the paper is nicely presented and definitely an interesting work. However, given the fact that methodologically we have not learned anything new from this paper and in combination with the not satisfying experimental evaluation I warrant for rejection.

---

> ### Author Response · Authors · 2018-11-14
> **Thank you for your detailed review.**
>
> To address your concerns, we have added the requested comparisons and clarifications to the paper. We also clarify several points about our methodology in this comment. We believe that these changes and our response address all issues, but we would appreciate any other feedback you might offer.
>
> We have included in Figure 9a a comparison to PPO (purple line) on the 2D navigation task as requested, and we will include the other simulated tasks as well in the final. This method performs well and converges to a better policy than our method and TRPO, though the improvement is marginal and our method is still several orders of magnitude more data efficient. Note that we do not assume access to the underlying state as in GPS (Levine 2016), and we have compared our method to LQR-FLM directly on images which was unable to learn the tasks. If LQR-FLM is unsuccessful, then GPS will be unsuccessful as the neural network policy is supervised by the LQR-FLM policies. We have now emphasized this point in Section 6.2.
>
> We have clarified various parts of the paper as suggested. Regarding the specific point on closed-form vs constrained LQR, we note that both of these procedures utilize a quadratic cost, thus there is not a significant implementation difference in terms of the constrained vs unconstrained procedures, and we have clarified this in Section 4.2. We have added citations for prior work that assumes access to a simple representation, in particular, Levine 2014, Deisenroth 2014, and Nagabandi 2018. We have also cited Deisenroth 2014 to elaborate on modeling bias, which is discussed in detail in that paper.
>
> In response to your concerns about using a constrained policy update, we further clarify and justify our policy update method in Section 4.2. We would appreciate if you can take a look at this section and see if it helps explain our modeling and policy learning choices. Finally, to address your concern about the presentation of the results, we justify why we choose to plot the reward rather than the final distance for the reacher task in Section 6.2.
>
> In response to “methodologically we have not learned anything new”, we believe that the main methodological contribution of this paper is to wed structured representation learning and local model-based RL into a more principled method compared to prior work, such that approximate inference within the model exactly enables the local model method we derive. These individual components are based on prior work, however they are extended to meet the needs of our overall method, and to our knowledge this combination is a novel contribution. This is combined with a demonstration that our method can solve real world robotics tasks such as block stacking from only image observations in about half an hour of interaction time, which again to our knowledge has not been done to the level of performance we demonstrate.
>
> We would appreciate it if you could take another look at our changes and additional results, and let us know if you would like to either revise your rating of the paper, or request additional changes that would alleviate your concerns.

---

> > ### Comment · AnonReviewer3 · 2018-11-23
> > **Effort to the right direction. Still needs improvement though**
> >
> > I appreciate the authors' effort to improve the manuscript.
> >
> > Let me first explain my comment regarding "methodologically we have not learned anything new". From what I understood by thoroughly reading the paper and the supplementary material, the theoretical section does not present any new method/technique. It is rather a collation of other ideas which of course, in order to make it work, requires a heavy understanding of the problem, a lot of engineering and some extensions to prior work (seem to be marginal). I want to clarify that I am perfectly OK with that. However, this issue, in combination with the evaluation on a limited number of tasks and the fact that in 1 out of 4 tasks the model is nowhere close to solving the problem makes the paper weak.
> >
> > Regarding the reacher experiment, specifically, I raised the same concerns as Reviewer 1 and I am not satisfied by the authors' reply. They authors have added the following piece in section 6.2:
> > "On the reacher task, we plot the reward function as defined by Gym since this is the standard metric used to evaluate performance on this task, and as shown by the videos on our project website, achieving high Gym reward correlates strongly with solving the task in terms of distance to the goal".
> > First of al, the fact that it is the standard metric does not make it the right one (the fact that you do not report it for the other tasks suggests that the authors do not believe it is the right metric either). It is definitely a metric that can hide many pathologies. As for the video, it is only 1s long and we cannot conclude much from it. However, what we saw is that the proposed approach is nowhere close to solving the task. It is just a free fall of the arm towards the goal, so no "strong correlation" between reward and solving the task, which brings us to my point that plotting the accumulated reward gives no information regarding the performance of the model. Since we see that the model fails on the reacher task, why have the authors not studied other similar gym environments? More importantly, why they do not comment on the reasons why the model cannot solve the task  but they rather try to argue that it achieves higher reward? In my opinion, further analysis is required.
> >
> > Apart from the experimental evaluation, I agree with Reviewer1 that presentation of the methodology requires more work. I apologise for not bringing this up in my first review but I share the same opinion as Reviewer 1 that
> > the important details like the cost-functions and the optimisation problem should be present in the main text and not buried in the appendix. I feel like sections 3 and 4 are difficult to follow unless you thoroughly study the appendix.
> >
> > Overall, I appreciate the authors' effort, and based on their response and their additional comparisons I am willing to change my score to a 5 but I still do not believe that it is good enough for publication.

---

> > > ### Author Response · Authors · 2018-11-28
> > > **Thank you for your response.**
> > >
> > > We have made further updates based on your comments, and we would appreciate any other feedback you might offer.
> > >
> > > As requested by both you and reviewer 1, we have moved details about the cost function learning and model optimization from the appendix to the main paper. We believe that this should clarify the presentation in the main paper, particularly sections 3 and 4, without relying as heavily on the appendix. Also as requested by both you and reviewer 1, the project website ( https://sites.google.com/view/iclr19solar ) now includes longer videos of the final policies from both our method and the comparisons for the simulated car and reacher tasks. We believe that the new video of the reacher policy learned by our method should demonstrate that are indeed learning the task. We perform worse than PPO, which solves the task nearly perfectly with about 40 times more data, however we significantly outperform the ablations and we perform comparably to TRPO. In the video, the policy we learn consistently brings the reacher arm close to the target in all ten episodes, and we would appreciate your re-evaluation of this experiment given the new videos.
> > >
> > > Regarding your concerns about the metric we use for reacher, we included the standard Gym reward function for this task because it is a standard Gym task, and this is the metric that prior works report. We could of course also report something else, but our evaluation metric is consistent with prior work in this regard. If the standardized reward function for this task (which we did not design) does not exhibit the desired behavior, this is the fault of the reward function, not the algorithm. Of course, we agree with the reviewer that reward doesn't tell the whole story, which is why we added three other tasks and included an extensive qualitative evaluation on the real robot task, which is the hardest task that we evaluate on. However, we do not believe that it is appropriate to criticize the work for using a standardized evaluation protocol, especially when we made an intentional effort to avoid the pitfalls of this evaluation by including other additional experiments, including an experiment with a real robot.
> > >
> > > We would like to emphasize the fact that our experimental evaluation includes four tasks, including one task on a real-world physical robot. All tasks involve learning directly from images. Compared to many recent works on model-based reinforcement learning that evaluate on four tasks [Watter ‘15, Banijamali ‘17, Nagabandi ‘18, Chua ‘18], the scope of the evaluation is comparable, and in contrast to all of these recent works, we evaluate on real-world image-based robotic control. We believe that it would be appropriate to evaluate the experimental evaluation comparatively to other works in the field, as this would seem to be the only fair standard against which to judge the work.
> > >
> > > We appreciate your responsiveness in helping us improve our paper, and we would appreciate it if you could take another look at our updates.

---

> ### Author Response · Authors · 2018-11-23
> **We have made some further updates to the paper and project website.**
>
> We have updated the paper with the requested comparisons, in particular, we have now included the results for PPO for all simulated tasks in Figure 4 and Appendix F. The project website (https://sites.google.com/view/iclr19solar ) also now includes a longer video of the learning process of our method on the Sawyer block stacking task. We would appreciate it if you could take another look at our changes and additional results.

---

### Official Review · AnonReviewer2 · 2018-11-06
**Interesting idea, but insufficient comparison to baselines**

**Rating:** 5
**Confidence:** 3

**Review:**

This paper proposes a model-based reinforcement learning approach, called SOLAR,
which consists of mapping complex, high-dimensional observations to low-dimensional
representations where transition dynamics between consecutive states are approximately linear.
In this low-dimensional space, local models can easily be fit in closed form and then used to optimize a policy, using a similar method to Guided Policy Search (GPS). The method is evaluated in 4 different settings (3 simulated, 1 on a real robot).

*Quality: the method seems to work well in the experiments. However, there are issues with the experimental evaluation (detailed below) which make it unclear whether the method is better than standard baselines.

*Clarity: the paper is well-written and clear overall.

*Originality: the paper proposes an extension of GPS, which to my knowledge is novel.

*Significance: the idea of learning representations where transitions are linear seems well-founded and potentially useful. However the merits of this method are not yet clear from the experiments.


Specific Comments:

- Please include an illustration of the 2D navigation task in Figure 3a
- I'm confused by the poor performance of E2C in the 2D navigation task.
The previous works of [Watter et. al, 2015] and [Banijalami et. al, 2017] report close to 100% accuracy using similar methods. Is the task formulated differently here?
- I would think a global action-conditional forward model (represented as convnet+deconvnet, and trained unrolled on its own predictions to reduce model errors) would perform quite well on the 2D navigation task, and possibly on the reacher task. Even though these are represented as images, they are very simple images with little distracting information, no changes in illumination/perspective, etc. It seems the model essentially just needs to learn a pixel translation for each action for the navigation task, and some rotations for the reacher. It already seems to work quite well for the non-holonomic car, which requires learning similar transformations. This baseline should be included for all the tasks.
- Although it does seem that the method performs well on the stacking tasks for the real robot, there are no baselines included. However, there are many works which have explored representation learning and control for robotics using neural networks. A couple examples (+see references within):

"Learning to poke by poking: Experiential learning of intuitive physics" Pulkit Agrawal, Ashvin V Nair, Pieter Abbeel, Jitendra Malik, Sergey Levine. NIPS 2016
"Deep Visual Foresight for Planning Robot Motion" Chelsea Finn, Sergey Levine ICRA 2017

At the very least, the method should be compared to pixel-based global models and representations learned with some kind of autoencoder or forward model for the robot task.

The paper proposes what seems to be a good idea, but it is not yet demonstrated by the current experiments.

---

> ### Author Response · Authors · 2018-11-14
> **Thank you for your detailed feedback and insightful comments.**
>
> We have addressed the issues you raised by adding the requested comparisons and evaluations. We also clarify the performance of the E2C comparison in this comment. We believe that these changes and our response address all issues, but we would appreciate any other feedback you might offer.
>
> As requested, we now have a comparison to a version of deep visual foresight, which is a pixel space global model, on the real robot block stacking task in Figure 5 (brown line). We will include a full and fair comparison to deep visual foresight on all tasks in the final, however, we note that the main difficulty here is data efficiency -- both deep visual foresight and Agrawal/Nair 2016 utilize weeks of robot data, whereas we demonstrate successful block stacking in about 30 minutes of interaction time. We believe that our method’s data efficiency in solving image-based tasks on a real robot is a significant result.
>
> We have also now included the other requested baselines on the block stacking task in Figure 5, specifically using an autoencoder representation (VAE ablation, pink line) and using a forward model (global model ablation, blue line). The ablations in this setting are competitive with our method, though our method still achieves a better final policy that is able to more consistently stack the block successfully. We also now compare against these baselines in Figure 4 for all simulated tasks as requested, and though they work well for 2D navigation, they are less successful on the reacher. Finally, as suggested, we include a diagram of the 2D navigation task in Appendix E, Figure 6.
>
> As you noted, E2C performs rather poorly on our version of the 2D navigation task. Our task is harder than that of Watter 2015 and Banijamali 2017, since the policy must reach every target position (which is appended to the state) rather than just a single position in the bottom right. In Appendix F.1, we show that E2C does perform better on the single-target task. Also note that we have used code directly from Banijamali and we have had discussions with these authors who noted similar challenges in reproducing the results of E2C. These works have not made their code publicly available, and to our knowledge no one has been able to reproduce their reported results. However, we are confident that our implementation is correct and that the E2C results accurately reflect the performance of that method.
>
> We would appreciate it if you could take another look at our changes and additional results, and let us know if you would like to either revise your rating of the paper, or request additional changes that would alleviate your concerns.

---

> ### Author Response · Authors · 2018-11-23
> **We have made some further updates to the paper and project website.**
>
> We have updated the paper with the requested comparisons, and the project website (https://sites.google.com/view/iclr19solar ) also now includes a longer video of the learning process of our method on the Sawyer block stacking task. We would appreciate it if you could take another look at our changes and additional results.

---

### Official Review · AnonReviewer1 · 2018-11-14
**Interesting model based RL approach that needs additional evaluation and clearer algorithm description**

**Rating:** 5
**Confidence:** 4

**Review:**

Summary:

The paper proposes SOLAR a model based RL algorithm that learns a low dimensional embedding such that the dynamics within the latent space are linear. Within this latent space the linear dynamics  are learned using a Bayesian regression. In addition, a quadratic cost function is approximated. The learned dynamics and the cost function are used to update the policy, while simultaneously bounding the change in policy by a KL-bound.  In contrast to other model-based RL algorithms the learned dynamics are not used for planning or imaginary roll-outs and are only used to improve the policy.

Review:
The introduction and experiment section is clearly written but the algorithm description lacks clarity and details, which hinder the understanding of the complete algorithm. One understands the motivation and the main approach but lacks a detailed understanding. For my personal taste the detailed description of learning the embedding is missing. I personally would prefer the statement of the cost-functions and the optimisation problem within the paper and not the appendix. The same holds true for the policy improvement. Therefore, I do not fully understand the approach without extensively studying the appendix or the references. Especially the contribution remains unclear. I am not aware how much the previous work had to be extended.

The experimental evaluation focuses on learning control signal to achieve certain trajectories, where the observations are high-dimensional images rather than low-dimensional representations. I personally think that these tasks are unnecessarily made more complex to incorporate high-dimensional images. Especially, the Sawyer experiment throws away all joint information even though the reward function is solely defined in joint/end-effector position. However, I am aware that this is general practice in the RL community. From the learning curves it seems that the approach is working and achieving good sample complexity compared to model free approaches. However, the improvement over the naive VAE approach remains unclear. I would like to see more comparisons to other model-based approaches. In addition, I am missing qualitative comparisons as the learning curves can be misleading. Especially, the videos on the homepage are really short and do not provide a good overview about the actual performance. Furthermore, you are not providing videos for all models in comparison. The 1s video of a single episode on the reacher task make me wonder what happens in the other episodes. Could you please add longer videos for all comparisons. Furthermore, it would be interesting how the trajectories evolve over time. Could you plot these trajectories?

Furthermore, it would be really interesting to try your approach on breakout. And test if your approach is learning the actual game dynamics and does not overfit to the block configuration.

Further minor comments:
- "This shifts our problem setting to that of a partially observed MDP, as we do not observe the latent state"
You are mentioning that you are solving a POMDP. Could you elaborate how you exploit the POMDP formulation and relate your work to POMDP algorithms. In addition, how do you define partial observability?

- You claim "our method is also successful at handling the complex, contact-rich dynamics of block stacking, which poses a significant challenge compared to the other contactfree tasks." I am quite doubt-full about the claim. Is the dynamics model really modelling contacts and is your policy really reacting to these contacts? Or is your policy just tying to follow a trajectory? From your current evaluation and the videos, I personally wouldn't conclude this. Could you elaborate how you come to this conclusion and provide additional evaluations to solidify your argument?

- You are not describing the action space for the Sawyer experiment. Are you using torques, velocities or positions? Can you guarantee that the control sequence is smooth? If not how do you ensure that the policy does not harm the robot?

- Could you please incorporate the exact  reward functions for each experiment within the appendix.

- Figure 4. Thanks a lot for including the additional model free baselines and adding all learning curves. However, the learning curves raise multiple questions:

(1) The Global Model Ablation, i.e. the MPC in latent space, works well in the the navigation and car experiment however fails
to achieve a meaning-full policy within the reacher task. Even though the initial performance is significantly better than
the other policies. Do you have an explanation for this failure?

(2) The LDS SVAE and VAE Solar version on the reacher task experiences jumps in performance even though the change between policies is bounded by a KL-Bound and the cost function is smooth. How do you explain these jumps? Furthermore, why are these jumps only occurring within the reacher tasks and not the other experiments.

(3) You are still missing the PPO baselines for the reacher and car experiment. Could you further explain the qualitative difference between the model-free and model-based policies. The difference in learning curves can be misleading.

(4) What is the unit of "Average Distance to Final Goal"? Is this measured in pixel or a different unit?

- Figure 5: You are plotting the distance to the goal as performance measure for the Sawyer experiment. The final policy has an approximate error of 2.5 cm. From just the learning curve I cannot conclude that the robot actually learns the task successfully. Is the block really stacked or can it also be wedged? Could you please provide image overlays of the last 10 episodes such that one can evaluate the qualitative performance?

---

> ### Author Response · Authors · 2018-11-23
> **Thank you for your detailed feedback.**
>
> We have added the requested comparisons and evaluations, and we hope to provide answers to your questions in this comment.
>
> As requested, we have updated the project website (https://sites.google.com/view/iclr19solar) with a longer video of the learning process for the Sawyer block stacking task with more example trajectories during training and from the final policy. Similarly to including image overlays of these trajectories, this should make clear that our method is indeed learning the task successfully. The video now depicts both the image observations we provide to the policy along with our model’s reconstructions of the images, which are generally fairly accurate. Also, as we discuss in Section 6.3, the video shows that the final policy successfully handles shifts in the position of the bottom block, and this indicates that the policy is reacting to different contacts rather than simply following a trajectory, as this strategy would fail more often. We will also update the website with videos depicting several episodes from all final policies for each method, which should help clarify our method’s performance compared to each of the baselines.
>
> As requested, we have added trajectories of the latent state for the 2D navigation task to Appendix F. We also clarify in Appendix E that we use velocity control for the Sawyer experiment, which helps ensure that the policy is safe along with an explicit safety constraint that clips velocities that are too high. We have added the exact reward functions for each task also to Appendix E. Finally, we have added all of the PPO comparisons to Appendix F, which should elaborate on the differences between the model-based and model-free policies.
>
> We clarify several points in the paper in response to some of your questions, including clarifying how we define partial observability in Section 2.2, explaining the global model failure on reacher in Section 6.2, and elaborating on the contacts in the block stacking task in Section 6.3. We now address the specific questions you raise:
>
> > “I would like to see more comparisons to other model-based approaches.”
>
> We are happy to include any comparisons to prior approaches that you can recommend, though we believe that our comparisons in the main paper and Appendix F already cover a substantial subset of the model-based approaches including LQR-FLM, MPC-NN, baselines similar to E2C and deep visual foresight, and several ablations. Note that LQR-FLM and MPC-NN were only successful from states rather than images as in Appendix F, so we subsequently added other comparisons and ablations as suggested by the other reviewers. Most prior model-based methods cannot operate directly on complex image observations in a data-efficient manner as we have demonstrated with our method, and we believe this is a significant result.
>
> > “it would be really interesting to try your approach on breakout.”
>
> Note that Breakout and other Atari games utilize discrete actions, and our model is designed for and tested on continuous action domains as we focus on robotic applications. Extending our model to discrete actions is non-trivial as it would necessitate some type of continuous relaxation or learned action representation, and we believe that this is an interesting direction for future work. We have added this point to Section 7, and we are happy to run evaluations on other tasks at your request for the final paper.
>
> > “How do you explain these jumps [in the reacher plot]?”
>
> The jumps occur whenever the policy is updated, which happens after we collect a batch of data. These jumps are also present in all of the other tasks, however they are less noticeable because of the greater variance in policy performance on those tasks.
>
> > “What is the unit of ‘Average Distance to Final Goal’”?
>
> This is measured as the ground truth distance in simulation, for which units are not meaningful. For reference, the x-y coordinates for both the 2D navigation and car tasks are between -3 and 3, so the final distances we achieve indicate that the agent essentially reaches the goal.
>
> We would appreciate it if you could take another look at our changes and additional results, and let us know if you would like to either revise your rating of the paper, or request additional changes that would alleviate your concerns.

---

> ### Author Response · Authors · 2018-11-28
> **We have made further updates to the paper and project website based on your feedback.**
>
> As requested, we have moved details about the cost function learning and model optimization from the appendix to the main paper. We believe that this should clarify the presentation in the main paper without relying as heavily on the appendix. Also as requested, the project website ( https://sites.google.com/view/iclr19solar ) now includes longer videos of the final policies from both our method and the comparisons for the simulated car and reacher tasks. This should help clarify that all methods are fairly consistent in their behavior by the end of training, and our method and the model-free algorithms are qualitatively more successful than the ablations.
>
> You and reviewer 3 have raised similar concerns about the quantitative results that we would like to address. For the reacher task, we included the standard Gym reward function for this task because it is a standard Gym task, and this is the metric that prior works report. We could of course also report something else, but our evaluation metric is consistent with prior work in this regard. If the standardized reward function for this task (which we did not design) does not exhibit the desired behavior, this is the fault of the reward function, not the algorithm. Of course, we agree with the reviewer that reward doesn't tell the whole story, which is why we added three other tasks and included an extensive qualitative evaluation on the real robot task, which is the hardest task that we evaluate on. All tasks involve learning directly from images. Compared to many recent works on model-based reinforcement learning that evaluate on four tasks [Watter ‘15, Banijamali ‘17, Nagabandi ‘18, Chua ‘18], the scope of the evaluation is comparable, and in contrast to all of these recent works, we evaluate on real-world image-based robotic control. We believe that it would be appropriate to evaluate the experimental evaluation comparatively to other works in the field, as this would seem to be the only fair standard against which to judge the work.
>
> We would appreciate it if you could take another look at our updates and let us know if you would like to revise your score or request further additions and clarifications.

---

> > ### Comment · AnonReviewer1 · 2018-12-11
> > **Thanks for the clarifications**
> >
> > The updated paper definitely goes in the right direction. I like that you updated the videos for all experiments.
> >
> > I think the videos interestingly highlight the current state of the art model based RL, which is - unfortunately - nowhere close to solving the task. This statement is not specific to the SOLAR algorithm and applies to most currently known model-based RL algorithms. In the non-holonomic car experiment PPO drives quite smoothly to the desired location while SOLAR learned that driving against the wall gets the agent closer to the target. Similar for the reacher task, where PPO goes smoothly to the desired location while SOLAR only goes in the direction to the target but not really close. Yes, SOLAR learns to increase the reward with a very good sample complexity - and does this comparable / slightly better than the other methods - but SOLAR is not close to solving the task and converges prematurely. Therefore, I am quite objecting these conclusions:
> >
> > "Not only is our method able to solve this task directly from raw, high-dimensional camera images within 200 episodes, corresponding to about half an hour of interaction time, our method is also successful at handling the complex, contact-rich dynamics of block stacking. As seen in the video on our project website, our method learns a policy that can react to slightly different contacts, due to the bottom block shifting between episodes, and is ultimately successful in stacking the block in most episodes."  => Well, yes the distance to the goal reduces with episodes and the final policy can sometimes stack the blocks. However, there are also episodes where the blocks are not completely stacked. Furthermore, there is no evaluation of different block positions.
> >
> > "The key insights in SOLAR involve learning latent representations where simple models are more accurate and utilizing PGM structure to infer dynamics from data conditioned on entire real-world trajectories. Our experimental results demonstrate that SOLAR is competitive in sample efficiency, while exhibiting superior final policy performance, compared to other model-based methods. " => Well, yes the performance might be slightly better but the paper provides no argument for simple models other than empirically shown on 3 experiments the simple models performed slightly better. The open questions are still, why is this converging prematurely and is this premature convergence easier to fix with global or local models?
> >
> > So concluding: The paper is ok written and does not contain major issues. The idea is not rather novel but a solid extension. For the solid extension, the results are not significantly better and lack general conclusions applicable to model-based RL. So I don't have a key take-away other than somebody has tried it and it resulted in mixed performance. So for me it is still borderline, i.e., a rating 5.5.

---

> > > ### Author Response · Authors · 2018-12-13
> > > **Thank you for your response.**
> > >
> > > We appreciate your feedback, however we disagree with several points in your assessment.
> > >
> > > We find your point on model-based RL in general to be unreasonable in judging the merit of our work. As our paper addresses model-based RL, our method of course will exhibit some of the limitations of the current state of the art in model-based RL while addressing other limitations. As you mention, it is well-known that model-based RL typically lacks the final performance reached by model-free RL. However, this criticism can be applied to any model-based RL research, and blanket rejection of all model-based RL research is unreasonable. Our aim is to expand the scope of domains that model-based RL can be applied to rather than improve the final performance of model-based RL in general. While the latter is an important line of work, it is not within the scope of our paper, and we believe that our method as is already demonstrates the novel ability to operate directly on image-based domains with orders of magnitude fewer samples than previously reported.
> > >
> > > This leads to our objection of your claim that our method is “nowhere close to solving the task”. Which video would support this claim? In the nonholonomic car case, the policy is successful at reaching the target at least 80% of the time, and furthermore it cannot be faulted for driving against the wall as this is not directly penalized by the cost function. For the reacher, while we agree that the behavior is noticeably worse than PPO, the final position of the end effector is consistently within a few pixels of the target, not “not really close” as you suggest. For the block stacking, even conservatively speaking, the final policy is successful on four out of five tries, and we note that even a carefully hand-engineered controller would not be 100% successful due to the inherent noise in controlling a real-world robot. Regarding your point on having multiple block positions, we have run this experiment multiple times and the block position has not been fixed between runs. Our method was successful every time at learning a good final policy, and we will formally include multiple block positions in the final version of our paper.
> > >
> > > As we have noted, we believe that the key take-away from our work is the development of a method that can operate in domains with complex observations such as images in an extremely data-efficient manner. This method strikes a favorable balance in extending the capabilities of model-based RL while being practical for use in real world tasks, which is generally not true for model-free RL algorithms. We again emphasize that, to the best of our knowledge, no prior work has shown the ability to solve a real world task as complex as block stacking with only half an hour of total interaction time, and we encourage you to either point to prior work that is comparable or reconsider your stance on the significance of this result.

---

### Meta-Review · Area_Chair1 · 2018-12-16
**Interesting ideas, but the paper can be improved.**

**Confidence:** 5
**Recommendation:** Reject

**Metareview:**

This paper proposes a method to learn representations to infer simple local models that can be used for policy improvement. All the reviewers agree that the paper has interesting ideas, but they found the main contribution to be a bit weak and the experiments to be insufficient.

Post rebuttal, the reviewers discussed extensively with each other and agreed that, given more work is done on a clear presentation and improving the experiments, this paper can be accepted. In its current form however, the paper is not ready to be accepted. I have recommended to reject this paper, but I will encourage the authors to resubmit after improving the work.